# A Review of Transition Metal Sulfides as Counter Electrodes for Dye-Sensitized and Quantum Dot-Sensitized Solar Cells

**DOI:** 10.3390/ma16072881

**Published:** 2023-04-04

**Authors:** Layla Haythoor Kharboot, Nor Akmal Fadil, Tuty Asma Abu Bakar, Abdillah Sani Mohd Najib, Norhuda Hidayah Nordin, Habibah Ghazali

**Affiliations:** 1Department of Materials, Manufacturing, and Industrial Engineering, Faculty of Mechanical Engineering, Universiti Teknologi Malaysia, Skudai 81310, Johor, Malaysia; kharboot@graduate.utm.my (L.H.K.); tuty@utm.my (T.A.A.B.); abdillahsani@utm.my (A.S.M.N.); 2Materials Research and Consultancy Group, Faculty of Mechanical Engineering, Universiti Teknologi Malaysia, Skudai 81310, Johor, Malaysia; 3Department of Manufacturing and Material Engineering, International Islamic University Malaysia, Jalan Gombak, Kuala Lumpur 53100, Selangor, Malaysia; norhudahidayah@iium.edu.my; 4College of Engineering and Science, Victoria University, Footscray Park Campus, Ballarat Road, Footscray, P.O. Box 14428, Melbourne, VIC 8001, Australia; habibah.ghazali@vu.edu.au

**Keywords:** dye-sensitized solar cell, quantum dot-sensitized solar cell, counter electrode, transition metal sulfide, polysulfide electrolyte, materials performance

## Abstract

Third-generation solar cells, including dye-sensitized solar cells (DSSCs) and quantum dot-sensitized solar cells (QDSSCs), have been associated with low-cost material requirements, simple fabrication processes, and mechanical robustness. Hence, counter electrodes (CEs) are a critical component for the functionality of these solar cells. Although platinum (Pt)-based CEs have been dominant in CE fabrication, they are costly and have limited market availability. Therefore, it is important to find alternative materials to overcome these issues. Transition metal chalcogenides (TMCs) and transition metal dichalcogenides (TMDs) have demonstrated capabilities as a more cost-effective alternative to Pt materials. This advantage has been attributed to their strong electrocatalytic activity, excellent thermal stability, tunability of bandgap energies, and variable crystalline morphologies. In this study, a comprehensive review of the major components and working principles of the DSSC and QDSSC are presented. In developing CEs for DSSCs and QDSSCs, various TMS materials synthesized through several techniques are thoroughly reviewed. The performance efficiencies of DSSCs and QDSSCs resulting from TMS-based CEs are subjected to in-depth comparative analysis with Pt-based CEs. Thus, the power conversion efficiency (PCE), fill factor (FF), short circuit current density (*J*_sc_) and open circuit voltage (*V*_oc_) are investigated. Based on this review, the PCEs for DSSCs and QDSSCs are found to range from 5.37 to 9.80% (I^−^/I_3_^−^ redox couple electrolyte) and 1.62 to 6.70% (S^−2^/S_x_^−^ electrolyte). This review seeks to navigate the future direction of TMS-based CEs towards the performance efficiency improvement of DSSCs and QDSSCs in the most cost-effective and environmentally friendly manner.

## 1. Introduction

Globally, researchers and scientists are particularly interested in meeting future energy requirements in parallel with rising energy demand and consumption [1,2]. Fossil fuels represent the primary global energy source. However, the energy generated from these fuels is not sustainable; fossil fuels deplete natural resources and are the primary source of greenhouse gas (GHG) emissions [3]. Thus, these GHG emissions can result in global warming, which is a perceived threat to human existence and survival on the earth. Consequently, new energy supplies are necessary to minimize or neutralize our energy dependence on fossil fuel resources [4].

Renewable energy is recognized as the most acceptable and long-term solution to meet the rising energy demand while being a source of long-term power generation. The European Union (EU) has proposed to increase a target at least 45% up from 32% renewable energy contributions to the energy consumption of the EU by 2030 [5]. Similarly, the United States has recorded an investment of over $90 billion in the technological development of clean energies [6]. For example, wind, sunlight, geothermal, hydropower, ocean waves and currents, biomass, and the temperature differential in the sea are under consideration as energy sources, due to the increasing usage and decreasing reserves of fossil fuels [3]. Following the predictions of several econometric models, fossil fuel reserves may be fully consumed by 2042 [7], thus emphasizing the importance of exploring renewable energy resources. Renewable energy has several advantages that promote overall environmental sustainability, including limiting the discharge of dangerous air pollutants and GHG emissions [8].

The energy from the sun provides exceptional environmental benefits compared with other forms of energy. Solar energy does not emit GHG or CO_2_, diminish natural resources, or generate waste matter [9,10]. Regarding solar radiation, 3.8 million EJ are produced annually, approximately 10,000 times greater than current energy demands [11]. Solar photovoltaics (PVs) are a very attractive option based on several renewable technologies. Solar PVs have been described as the most widely accepted solar-to-electricity technology [12,13]. The construction of the first solar cell was recorded in Bell’s lab in 1954, followed by the emergence of various types of solar cells [14]. Solar PVs are categorized into silicon-based (first-generation), thin-film-based (second-generation), and the current emerging third-generation solar cells [15]. The first- and second-generation solar cells have recorded dominance in the PV market, with an estimated total market share of over 95% [13]. First-generation solar cells are based on single or multi-crystalline *p*-*n* junction silicon materials, with a PCE of over 20% [16]. Nevertheless, the high purity requirements of silicon crystals, high -temperature fabrication requirements, expensive fabrication materials, and sophisticated processing techniques associated with negative impacts on the environment are of great concern in their global applications [17,18,19,20,21,22]. In second-generation PVs, thin film-based solar cells are categorized into cadmium telluride (CdTe), copper indium gallium selenide (CIGS), and amorphous silicon (a-Si) [23]. These second-generation PV devices are cheaper but less efficient than first-generation solar cells [22,24] While the efficiency of third-generation solar cells is rated greater than that of thin film solar cells, it is still less than in first-generation solar cells [25]. Third-generation solar cells consist mainly of dye-sensitized solar cells (DSSCs), organic solar cells, perovskite solar cells, and quantum dot-sensitized solar cells (QDSSCs) [13,26]. Kouhnavard et al. [22] demonstrated that high-efficiency devices with low production costs were possible with the advancement of third-generation solar cells, such as DSSCs, QDSSCs, and organic solar cells.

The emerging third-generation solar cells are still in the production process for commercialization. Thus, DSSC has gained significant attention among third-generation solar cells associated with low production cost, low light requirement, and mechanical robustness [21,27]. Oregan and Gratzel initially developed the first DSSC with a 7.1% photoelectric conversion rate [21,28]. Hence, extensive efforts were emphasized to increase the efficiency of DSSCs. Such effort is yet to achieve greater than 12%, the highest recorded efficiency over the last 10 years [22,29,30]. DSSCs and QDSSCs were significant forms of third-generation solar cells developed during the last two decades, and discovered to operate efficiently in indoor environments: They are still being subjected to further technological improvement to maximize their efficiency [28,31]. These cells produced high efficiency in a wide range of illumination wavelengths, including LED and indoor fluorescent tubes. Furthermore, the QDSSC is a simple homolog of the DSSC. The only noticeable difference is that the organic or organometallic dyes are replaced with quantum dot (QD) sensitizers, such as CdSe, CdS, PbSe, PbS, and InP [32].

Previous review articles have reported on the current performance of DSSCs and QDSSC in general without focusing on either transition metal chalcogenides (TMCs) or transition metal dichalcogenides (TMDs) [33]. In this review, the current performance of electrochemical and photovoltaic properties of low-cost catalytic CEs developed from earth-abundant TMCs, including TMDs and their composites with other materials, are discussed: This is the novelty of this review paper.

## 2. DSSCs

The DSSC is a semiconductor PV device directly converting solar radiation to electric current for intended final consumption. Therefore, the major components of a DSSC are reported by Gong et al. [34] as follows:A working electrode consists of a transparent conductive oxide (TCO) glass substrate sheet treated with a mesoporous oxide layer to activate electronic conduction.Molecular dye covalently bonded to the TCO for enhancement of light absorption.Redox mediator-based electrolyte to enable regeneration of oxidized dye molecules.Cathode electrodes consisting of TCO are mainly coated with platinum (Pt) to facilitate the collection of electrons.

In achieving high power conversion efficiency (PCE) values, several studies proved that each component of DSSCs functioned collectively towards the overall cell performance. Therefore, the high PCE advantage for DSSCs under visible indoor light or low illumination was a possible alternative to traditional electric power sources for portable electronics and devices that operate under ambient light conditions [35]. A typical representation of DSSC is illustrated in Figure 1, which depicts its major components and the oxidation-reduction states of the electrolyte towards molecular dye regeneration [36].

Sharma et al. [37] reported that a TCO glass substrate in a DSSC should produce high transparency (>80%) and equally be of high electrical conductivity. These properties would allow optimum sunlight penetration, efficient charge transfer, and reduced energy losses in DSSCs. Therefore, the TCOs in DSSCs could be a fluorine-doped tin oxide (FTO) or indium-doped tin oxide (ITO) substrate. Concurrently, the depositing thin layer of semiconducting material could be any of TiO_2_, Nb_2_O_5_, ZnO, SnO_2_ (*n*-type), SrTiO_3_, CeO_3_, or NiO (*p*-type). Bavarian et al. [38] discovered that fluorine-doped tin oxide (SnO_2_:F) was among the most widely used TCO substrates. For wide bandgap semiconducting material oxides acting as sensitizers, nanoparticle-based TiO_2_ was the most preferred material in DSSCs [21,39].

TiO_2_ nanostructures can be synthesized in several ways: sol-gel, hydrothermal, microwave, sonochemical, solvothermal, flame spray pyrolysis, anodization, direct oxidation, and the micelle method [21]. TiO_2_ occurs naturally in different anatase crystalline forms, associated with an efficient charge transport phenomenon and higher DSSC suitability than other TiO_2_ crystalline forms [21,39]. Such forms include rutile (tetragonal) and brookite (orthorhombic) phases [21]. In several studies, Tennakone et al. [40], Sayama et al. [41], and Sharma et al. [37] found a wider application of the TiO_2_ anatase allotropic form owing to the higher energy band gap (3.2 eV). Alternatively, the TiO_2_ rutile form was associated with an energy band gap of 3.0 eV. Figure 2 represents the energy band positions of several semiconductors. Notably, ZnO is a promising alternative to TiO_2,_ given the similarity of their energy band structure and relatively high electron mobility (1–5 cm^2^ V^−1^ s^−1^) [34,42].

Many artificial dyes have been synthesized, including commercial N3, N719, and Z907. Since the introduction of DSSC, these dyes have been responsible for the maximum absorption of incident light [34]. Sharma et al. [37] summarized that the dye used in DSSCs should be luminescent, consisting of ultraviolet-visible (UV-vis) and near-infrared (NIR) regions in the absorption spectra. Meanwhile, Sugathan et al. [43] and Sharma et al. [37] reported that the highest occupied molecular orbital (HOMO) of the dye should be significantly distanced from the TiO_2_ conduction band (CB) surface and lower than the redox electrolyte medium for successful regeneration of oxidized dye. Similarly, the lowest unoccupied molecular orbital (LUMO) should strongly adhere to the TiO_2_ surface and be higher than the TiO_2_ CB potential for efficient charge injection [21,37,43].

Dye aggregation on the surface of TiO_2_ led to the insertion of a co-absorbent, such as chenodeoxycholic acid (CDCA), phosphoric acid, and carboxylic acid groups, between the dye and TiO_2_ [37,44]. This process resulted in the prevention of aggregation and subsequent bridging of the likelihood of a recombination reaction between redox electrolytes and electrons of TiO_2_ nanolayers [37,44]. Another key component in all DSSCs is the electrolyte. Thus, intensive research has been conducted on each electrolyte constituent, including solvents, redox couples, and additives [44,45,46,47]. The electrolyte functions as a charge carrier, collecting electrons at the cathode and discharging them to the dye molecule. Regarding cell efficiency, the iodide/triiodide liquid electrolyte has been widely used due to its electrochemical kinetics [34].

In DSSCs, the counter electrode (CE) plays a critical role in catching electrons from the external circuit and accelerating the redox electrolytic reduction. Typically, the CE is synthesized by applying a thin catalyst layer on the conductive substrate. The essential criteria for CE materials are strong electrocatalytic activity, low resistivity for charge transmission, and long-time stability [47]. Noble metals, such as Pt, Ag, and Au, are the most attractive CE materials because of their great electrocatalytic activity for reducing redox couples in aqueous electrolytes or efficient hole transmission in solid-state electrolytes [48]. Subsequently, this reduction process regenerates the electrolyte. Alternatively, a catalyst is required to enhance the reaction kinetics of this process, and the choice depends on the ultimate application and material cost; noble metals are expensive, and their deterioration in a liquid electrolyte remains a concern. Therefore, transition metal sulfide (TMS)-based materials are potential alternatives to noble metal-based CE catalysts, and these are investigated and presented in this review.

The schematic representation in Figure 3 represents the working principle of the DSSC. The incident radiation leads to the excitation of dye molecules with HOMO electrons travelling towards the region of LUMO. From this point on, the electron travels towards the lowest energy level of the associated semiconductor CB. The electrons flow through the mesoporous surface of the semiconductor and towards the conduction electrode. The electrons then pass through the external load and accumulate at the CE, where electrolyte reduction occurs. Subsequently, the dye runs out of the electrons during this stage, which the electrolyte compensates for. Finally, the circuit loop is closed in this manner, and the current flow continues. This process is a forward charge transport mechanism. The efficiency of the DSSC can also be significantly impacted by reverse charge transfer (electron mobility).

The overall DSSC performance can be evaluated with several parameters as follows:Incident photon to current conversion efficiency (IPCE)Short circuit current (*I*_sc_)Open circuit voltage (*V*_oc_)Maximum power output (*P*_max_), which is a product of maximum voltage (*V*_max_) and maximum current (*I*_max_)Overall efficiency (*η*), which represents the percentage of solar energy converted into electrical energyFill factor (FF) at a constant light level exposure [37,49]

Some of these parameters are diagrammatically represented in a current density-voltage plot in Figure 4. The *V*_oc_ is expressed in Equations (1) and (2), where cell terminal voltage (*V*_t_), open circuit current (*I*_o_) represent an expression for short circuit current [25].
(1)$$V_{\mathrm{oc}}=V_{t}\mathrm{In}\left[\left(\frac{I_{\mathrm{sc}}}{I_{o}}\right)+1\right]$$
(2)$$I_{\mathrm{sc}}=I+{\;I}_{o}\left[\exp\left(\frac{V}{V_{t}}\right)-1\right]$$

The FF is expressed in Equation (3), with the maximum theoretical FF value being 1.0. Nonetheless, FF is limited to 0.83 owing to diode functionality limitations [50] and serves as an input in determining the *η* value for DSSCs in Equation (4). The division of electrical power density defines *η* by the incident solar power density (*P*_inc_) [25,50]. Moreover, *P*_inc_ is standardized at 1000 W m^−2^ for PV cells subject to testing at a spectral intensity equal to the intensity of the sun on the surface of the earth at an angle of 48.2° [50].
(3)$$\mathrm{FF}=\frac{V_{\max}{\;*\;I}_{\max}}{V_{\mathrm{oc}\;}{*\;I}_{\mathrm{sc}}}$$
(4)$$\eta =\frac{{\mathrm{FF}\;*\;V}_{\mathrm{oc}}{\;*\;I}_{\mathrm{sc}}}{P_{\mathrm{inc}}}$$

## 3. QDSSC

The QDSSC is part of the third-generation solar cell, while QD is used to replace dye due to its excellent optoelectronic properties [50,51,52]. QDs are nano-sized semiconductor particles with size-dependent physical and chemical properties. The notable characteristics of QDs include the tunability of energy band gaps, narrow emission spectrum, wide excitation spectra, good photostability, high molar extinction coefficient, and multiple exciton generation (MEG) [50,53,54]. Based on these advantages, the fabrication of QDSSC recorded efficiency of up to 7% [50,55,56] and PCE of up to 12.75%[57].

QDs are commonly cadmium chalcogenide (CdX), where X represents any of the elements S, Se, or Te, and produces CdS, CdSe, or CdTe. Other applicable QDs include CuInS_2_, PbS, AgInSe_2_, PbSeS, Ag_2_Se, and ZnS. Nevertheless, CdS and CdSe QDs have been considered stable materials for QDSSCs [22]. The major difference between DSSC and QDSSC was the replacement of dye by inorganic QD nanoparticles with the mesoporous TiO_2_ coated with QDs through in-situ fabrication or colloidal QD deposition (ex-situ fabrication) [51,58,59,60].

A typical representation of QDSSC and its operational principle is portrayed in Figure 5 [13]. The operational principle of the QDSSC initiates with its irradiation under sunlight. This process causes sunlight absorption by the QD sensitizers and generation of electron-hole pairs, in which the electrons become excited from the valence to the CB of the QDs. The electrons are injected from the QDs into the CB of TiO_2_ mesoporous films for onward transportation from the working electrode (photoanode). Subsequently, the electrons pass through the external circuit to the CE. Following the catalytic effect of the CE and the redox-couple effect of the electrolyte, the transfer of electrons occurs for the regeneration of oxidized QDs into their original ground state [13,53]. 

## 4. Transition Metal Chalcogenides (TMCs) Compounds-Based CE Catalysts

Transition metal chalcogenides (TMCs) are elements with partially filled *d* orbitals. Chalcogenide is a chemical compound containing at least one chalcogen anion and another electropositive element. Although all group 16 elements are classified as chalcogens, the term is usually used to refer to sulfides, selenides, tellurides, and polonides. The latter, typically semiconducting, have many redox sites, unusual crystal structures, high electrical conductivity, and excellent electrochemical capabilities, since many have a layered 2D structure. Additionally, TMC exhibits excellent thermal stability [61], efficient optical absorption due to its tunable indirect bandgap energies (1–2 eV), high absorption coefficients (105–106 1/cm), and unique physiochemical properties that enable the catalysts to absorb visible light (abundant in solar radiation).

TMCs have garnered considerable research attention for use in lithium-ion batteries, solar cells, hydrogen evolution, and fuel cells. Furthermore, the QDS of metal chalcogenides and nanostructures displays enhanced edge effects as the quantum confinement effect enables the use of TMC nanostructures under solar-simulated irradiation. Therefore, numerous researchers have improved their research using TMCs as precious metal substitutes in catalyst materials for energy storage and generation applications [62,63,64,65,66]. Although Pt is commonly recognized as the most efficient catalyst owing to its high electrical conductivity and activity [67,68], it is still expensive to produce [69,70]. Different alternative types of materials have been utilized as CEs, such as metallic compounds or composites [71,72,73], various forms of carbon [74,75], and conductive polymeric materials [76].

Several transition metal compounds have been used to substitute costly noble metal catalysts in the domains of hydro oxidation, hydrodesulfurization, and methanol oxidation [77,78,79]. Researchers continue to develop and improve electrode materials to suit the requirements of energy storage devices and generation systems while safeguarding the environment and reducing fossil fuel use through nanotechnology and renewable energy sources [80,81,82,83,84]. Consequently, significant efforts have been made to develop non-precious metal catalysts, such as available transition metals (metal carbides, sulfides, oxides, and nitrides), which are unique energy storage materials [62,63,85,86,87].

In DSSCs and QDSSCs, the development of transition metal compound catalysts to replace the costly Pt CE began in 2009 with TiN and CoS [88,89]. Other researchers, such as Quy et al. [90] and Sun et al. [91], discovered Ni_3_S_4_/FTO and NiS/FTO electro-deposited CEs in DSSCs and QDSSCs, respectively. The devices exhibited remarkable electrocatalytic activity in S^−2^/S_x_^−2^ and I^−^/I_3_^−^ redox-couple electrolyte systems with excellent electrochemical stability. Thus, their studies indicated NiS as a highly interesting candidate to replace Pt in photoelectrochemical cells employing I^−^/I_3_^−^. In the case of QDSSCs, the polysulfide electrolyte caused chemical adsorption and the corrosion of Pt CE.

TMSs are efficient materials used in most energy storage applications due to their excellent electrochemical characteristics. Moreover, TMSs can facilitate electron transfer in the structure of sulfides due to the small electronegativity value for S metals [92]. Zhao et al. [93] demonstrated high catalytic activity, amplified by metal nanoparticles with a high ratio between the surface area and particle volume as the catalytic process was located on the surface. Another study by Theerthagiri et al. [94] investigated the excellent properties of TMSs related to the sulfur-specific morphology of their surfaces in terms of unique shapes (nanosheet, nanorod, nanoplate, nanobud, and nanowires). Various types of TMSs, such as NiS, CuS, CoS, MoS_2_, and WS, were considered interesting compounds for CEs in DSSCs and QDSSCs [95,96]. Additionally, TMSs were ideal for cost-effective and Pt-free CEs due to their variable crystalline morphologies, adjustable stoichiometry, and improved catalytic performance, thus establishing them as an appealing technology for large-scale production.

TMSs were recently recognized as highly beneficial CEs compared to binary TMSs. The coexistence of two different cations allows for fascinating morphological characteristics, rich redox responses, controllable bandgap formation, and optical and electronic properties through modifying the proportions of their composition [97]. Therefore, this review article focuses on the performance of TMSs as CEs for DSSCs and QDSSCs.

Several studies have confirmed the ability to control the shape and size of TMS structures at the nanoscale (<100 nm) level, which could determine their design, characterization, production, and application [98,99,100,101,102]. Parveen et al. [103] described that the parameters and conditions of the response rate affected the morphological characteristics, shape, and size of the growing nanostructured materials. When the reaction rate was high, these characteristics grew anisotropic. Conversely, the materials grew isotropic when the reaction rate was low, as the reaction occurred under the control of thermodynamic conditions.

Jeevanandam et al. [104] demonstrated the typical use of nanostructures for numerous purposes, such as mechanical stability, increased visible light, reflection of harmful ultraviolet waves, and absorption of radiation. The study added that in the encapsulation process, the reactive nano-entities were encapsulated by non-reactive species to provide stability to the nanostructures. Therefore, it was essential to achieve high system performance in designing the surface area of materials while controlling the nanostructures and surface functionalization [105]. Mourdikoudis et al. [106] asserted the basic results of the physics and chemistry of solids, in which the most solid properties depended on the microstructure, such as chemical composition, arrangement of the atoms (the atomic structure), and solid form (1D, 2D, and 3D). The next subsection presents a detailed review of TMS-based CEs used in DSSCs and QDSSCs.

### 4.1. The TMS-Based CEs Applications in DSSCs

In TMS-based CE applications for DSSCs, Wu et al. [107] reported that laminar WS_2_ and MoS_2_ were synthesized upon adopting a simple chemical method as CEs for DSSCs. The study observed that WS_2_ and MoS_2_ performed well for triiodide reduction, with the DSSCs producing PCE values of 7.59 (MoS_2_) and 7.73% (WS_2_). These values were equivalent to the outcomes of Pt CE-based DSSCs (7.64%). Nonetheless, MoS_2_ and WS_2_ CEs yielded higher FFs of 0.73 and 0.70, respectively. These FFs demonstrated the associated high catalytic activity for triiodide reduction, in comparison to the FF of 0.66 produced by Pt CE an. The *V*_oc_ and short circuit current density (*J*_sc_) were relatively high respectively at 0.76 V and 13.84 mA cm^−2^ for MoS_2_-DSSC and 0.78 V and 14.13 mA cm^−2^ for WS_2_-DSSC.

In another study, He et al. [108] synthesized Ag_2_S nanoparticles as a CE catalyst. The colloidal synthesis approach was adopted, in which the synthesized Ag_2_S was on FTO glass. As demonstrated in Figure 6, the XRD pattern reveals that the obtained products are acanthite Ag_2_S (see Figure 6a). The TEM image also reveals the dominant presence of monodispersed nanocrystals, with a size of approximately 18 nm (see Figure 6b) [108]. DSSCs utilizing Ag_2_S CE revealed PCE of 8.40%, which was higher than the DSSC with Pt CE (8.11%). Moreover, the authors discovered that thickness significantly impacted catalytic activity on variation of Ag_2_S thicknesses from 0.11–1.05 μm, in which 0.53 μm was the ideal thickness for the CE. DSSCs with Ag_2_S CE at this thickness were associated with *J*_sc_, *V*_oc_, and FF values of 16.79 mA cm^−2^, 757 mV, and 0.66, respectively. Furthermore, the electrochemical test revealed that the Ag_2_S electrode produced a reduced charge transfer resistance (*R*_ct_) and improved electrochemical stability.

A study by Zhang et al. [109] developed a CuS nanosheet (CuS NS) network on a flexible substrate of polyethylene terephthalate (PET) as the CE. The CuS nanosheet networks acted as electron collectors and redox-couple catalysts. When the transmission reached 80%, the CuS nanosheet networks demonstrated good conductivity, with a sheet resistance of 20 Ω. The CuS nanosheet networks produced strong catalytic activity, with the DSSCs revealing a PCE of 6.38% (14% improvement over Pt CE-fabricated devices). Furthermore, DSSCs with CuS NS revealed *J*_sc_, *V*_oc_, and FF of 18.10 mA cm^−2^, 0.66 V, and 0.53, respectively. Alternatively, for Pt CE-DSSC, *J*_sc_, *V*_oc_, and FF were 15.81 mA cm^−2^, 0.70 V, and 0.506, respectively. After 100 bending and relaxing cycles, the efficiency of the bending tests decreased by 10%, thus demonstrating high mechanical stability. Hence, the cost of manufacturing DSSCs utilizing CuS nanosheet networks CE was significantly reduced owing to the absence of expensive Pt and FTO substrates.

Sun et al. [110] employed a facile process requiring the combination of a hydrothermal technique and post-annealing treatment for the synthesis and onward deposition of Sb_2_S_3_ on an FTO conductive substrate. According to the electrochemical characterization, the prepared Sb_2_S_3_ film revealed good electrocatalytic stability and activity for catalyzing the triiodide reduction. The PCE of DSSCs utilizing Sb_2_S_3_ CE, resulting from an Sb_2_S_3_ growth extension time of 24 h at 150 °C, was 5.37%. Therefore, the PCE was comparable to Pt CE-DSSC (5.36%). The Sb_2_S_3_ CE in DSSC was associated with *J*_sc_, *V*_oc_, and FF of 14.5 mA cm^−2^, 0.70 V, and 0.528, respectively. Additionally, the PV parameters for platinized CE in DSSC produced *J*_sc_, *V*_oc_, and FF of 12.5 mA cm^−2^, 0.65 V, and 0.653, respectively. Yue et al.’s study [111] synthesized VS_2_ decorated with carbon nanotubes (CNTs), which resulted in CNTs/VS_2_ CE through in situ hydrothermal treatment at 180 °C. The CNTs/VS_2_ CE in the DSSC was associated with *J*_sc_, *V*_oc_, and FF of 15.57 mA cm^−2^, 0.755 V, and 0.682, respectively. These values produced a PCE value of 8.02%, while the DSSC with platinized CE recorded a PCE value of 6.49% (*J*_sc_, *V*_oc_, FF of 14.03 mA cm^−2^, 0.717 V, and 0.645, respectively).

Bai et al. [112] developed a semi-transparent SnS_2_ nanosheet (NS) films (SnS_2_ NS) with a resultant thickness of about 300 nm using an environmentally friendly solution-processed approach. This material was used as a low-cost CE for triiodide reduction in DSSCs. The resulting SnS_2_ CE demonstrated high activity as a catalyst compared with high-cost Pt CE. The DSSC obtained through SnS_2_ NS CE produced a PCE of 7.64%. Furthermore, the SnS_2_ NS functionalized with a small amount of carbon nanoparticles produced a PCE of 8.06%, which was higher than the PCE (7.71%) of Pt CE-DSSC. Figure 7a,b illustrate the respective TEM and HRTEM images of the synthesized SnS_2_ nanosheets. The images revealed nanocrystals with lateral sizes of about 20–30 nm (see Figure 7a) and NSs comprising a few stacks of SnS_2_ single layers (see Figure 7b). Thus, this study concluded that SnS2 NS CE was appropriate for large-scale production of DSSCs as it was a simple construction process, low-cost, and highly transparent, and had good catalytic activity. Similarly, Yang et al. [113] discovered that SnS2 could replace Pt in DSSCs. DSSCs utilizing SnS2 CE produced good PCE (6.30%) following the adjustment of preparation conditions, thus demonstrating similar catalytic activity to Pt-based CE.

Another CE in DSSCs was based on the laminar-shaped Co_3_S_4_ nanosheets synthesized and deposited on an FTO glass substrate using a one-pot hydrothermal technique. The catalytic activity of Co_3_S_4_ nanosheets towards the iodide redox pair was observed to be exceptional. The PCE of the DSSCs with Co_3_S_4_ CE was 7.19%, equivalent to the PCE with Pt CE (7.27%). Therefore, the researchers concluded that the effective performance of Co_3_S_4_ nanosheets resulted from their unique “laminar-like” structure, which facilitated catalysis by providing a greater surface area for mass and electron transports [114]. Jin and He [115] utilized a hydrothermal approach to make monodispersed CoS_2_ nanocrystals (NCs), which were turned into nano ink for electrode fabrication by employing a simple cast-coating technique. In DSSCs, the CoS_2_ electrode demonstrated strong electrocatalytic activity towards the iodide redox pair. The highest PCE of the DSSC utilizing CoS_2_ nanocrystal as CE was 6.78%, which was comparable to the PCE (7.38%) for DSSCs with Pt CE. Additionally, the *J*_sc_, *V*_oc_, and FF for the DSSC with CoS_2_ NCs were 14.62 mA cm^−2^, 0.71 V, and 0.64, respectively, while the DSSC with Pt CE produced 14.78 mA cm^−2^, 0.72 V, and 0.68, respectively.

Huo et al. [116] produced electrophoretic deposition (EPD) and ion exchange deposition (IED) for the CoS layer placement onto the FTO glass substrate. The sulfide film was treated with aqueous solutions of NaBH_4_ and H_2_SO_4_, and the effect of these treatments on CoS catalytic activity was examined through the engagement of field emission scanning electron microscopy (FESEM), cyclic voltammetry, electrochemical impedance spectroscopy, and Tafel measurements. The CoS CE treated with H_2_SO_4_ and NaBH_4_ solutions produced good results. Thus, the DSSC utilizing CoS CE demonstrated a PCE of 7.72%. The internal structure and surface morphologies of the CoS film were modified by H_2_SO_4_ and NaBH_4_ aqueous solution treatments, revealing a honeycomb-like morphology with many folds and holes, which was necessary for efficient mass transport, electron transfer, and high catalytic activity.

Wang et al. [60] synthesized 2-D hexagonal FeS with high energy facets (HEF) (FeS-HEF) through the deployment of a solution-phase chemical technique and used FeS-HEF as a CE catalyst in the DSSC. As observed in the Tafel polarization and cyclic voltammetry studies, facets were critical for increasing the catalytic performance of iodide redox pairs. The FeS-HEF CE-based DSSC revealed a good PCE value (8.88%), which was about 1.15 times greater than the PCE (7.73%) of Pt-based DSSC. Meanwhile, Shukla et al. [117] used pyrolysis of thiourea and ferric chloride to make FeS_2_ films on an FTO glass substrate as CE for DSSC, with a high PCE (7.97%), superior to the Pt CE-based cell. The enhanced catalytic activity of FeS_2_ was credited with the improved overall efficiency of the device. This greater efficiency occurred as FeS_2_ was paired with strong optical properties and improved light dispersion in the solar cell.

Zhang et al. [118] reported FeS nanorods by electrospinning an iron (III) PAN/nitrate solution and then sulfurizing to synthesize CE with a PCE of 6.47%. This enhanced catalytic activity was attributed to the larger number of electron-hole pairs, iron sulfide’s electrical conductivity, and the mixed valence of Fe in iron sulfide. Thus, these advantages assisted the transfer of charges at the electrolyte/electrode interface. Furthermore, FeS nanorods outperformed Pt in mechanical strength because of their linked conductive channels and exceptional mechanical stability due to one-dimensional morphology. Another study, by Raj et al. [119], synthesized the MoS_2_ layer on FTO glass using the chemical vapor deposition method. This type of MoS_2_ CE produced high reflectivity, which rendered photon collection easier, thus resulting in a higher current density. Consequently, a PCE value (7.5%) greater than that of the Pt CE-based cell (7.28%) was observed.

Huang et al. [120] developed TCO using a simple solution approach to synthesize MoS_2_ on graphite paper (GP). The PCE of the DSSCs using GP/MoS_2_ CE (6.48%) was greater than that of the device using FTO CE/Pt (6.22%). The high catalytic activity of GP/MoS_2_ compared with FTO/Pt was due to the high conductivity of the GP substrate, effective electrical path between the GP substrate and MoS_2_ film owing to strong mechanical adhesion, and comparable *R*_ct_ of GP/MoS_2_ to FTO/Pt. Moreover, the GP/MoS_2_ electrode was very stable due to the strong crystallinity of MoS_2_ and the fact that it was securely fixed to the GP substrate. Meanwhile, Jeong et al. [121] presented an efficient MoS_2_ CE produced using a low-temperature technique (70 °C), followed by near-infrared laser sintering. The laser-sintered CE produced greater connectivity and crystallinity between the MoS_2_ nanoparticles than the heat-sintered MoS_2_ CE, thus resulting in strong catalytic activity for the iodide redox pair. Additionally, the PCE of laser-sintered MoS_2_ CE-based DSSC was 7.19%, higher than the device based on heat-sintered CE.

Zhang et al. [122] developed a transparent MoS_2_ film with a few atomic layers utilized as a CE for DSSCs. An artificial technique was used for forming active edge sites by hole patterning on MoS_2_ atomic layers to boost the MoS_2′_s electrode activity. The EIS analysis revealed that the performance of the catalyst was greatly improved after hole patterning. The DSSCs reported 2 and 5.8% PCE values pre- and post-hole patterning, respectively. A study by Li et al. [123] synthesized aligned NiS nanotube arrays and deposited them onto the FTO glass substrate for utilization as CEs for DSSCs. The DSSC with the NiS nanotube arrays demonstrated a PCE of 9.8%, greater than the cell that employed Pt and NiS nanoparticle CEs. The improved catalytic activity of orientated NiS nanotube arrays assisted electron transport in the axial direction with substrate NiS arrays. Additionally, Wu et al. [107] reported that NiS was an effective catalyst for CE in DSSCs.

Wan et al. [124] used a hydrothermal technique to manufacture hierarchical hollow NiS_2_ microspheres on FTO glass as CE. Several hollow NiS_2_ microparticle shells were partially fractured in SEM images, signifying strong catalytic activity and increased electrocatalytically active sites and electrolyte adsorption. The PCE for DSSCs based on hollow microsphere NiS_2_ CE was up to 7.84% compared with the PCE for DSSCs depending on Pt CE (7.89%). Yang et al. [125] also discovered a hydrothermal approach to produce NiS hollow spheres, with the hollow structure demonstrating more electrolyte absorption sites. Cells based on hollow NiS sphere CE produced a PCE of 6.90%, equivalent to that of Pt CE-based DSSCs (6.75%).

Table 1 summarizes the different TMS materials synthesized and deposited onto material substrate layers using diverse techniques by various authors. These techniques included colloidal synthesis, electrospinning, hydrothermal, post-annealing treatment, electrophoretic deposition, ion exchange deposition, solution-phase approach, spray pyrolysis, chemical vapor deposition, heat- and laser-sintered treatments, and solid-state sulfurization. The PCEs of DSSCs with these TMS-based CEs ranged from 5.37 to 9.80%, with I^−^/I_3_^−^ redox-couple electrolytes, mainly used in the regeneration of oxidized dye molecules. Other PV parameters, such as FF, *V*_oc_, and *J*_sc_, associated with the various DSSCs, are also defined.

Efficiency in the context of DSSCs refers to the amount of electrical power output that can be generated by the cell from a given amount of light input. The efficiency of the counter electrode depends on its catalytic activity and the rate at which it can facilitate the reduction of the electrolyte. A more efficient counter electrode will have higher catalytic activity, which means it can facilitate the reduction of the electrolyte more quickly, and therefore, generate more electrical power output. Platinum counter electrodes generally have higher catalytic activity and stability than transition metal counter electrodes, resulting in higher DSSC efficiencies. However, with the proper selection of transition metals and optimization of their properties, transition metal counter electrodes can also achieve high efficiencies, comparable to or even higher than Pt. Moreover, the lower cost and greater availability of transition metals make them attractive alternatives to Pt in DSSC applications, as shown in Table 1.

As presented in Figure 8, the relative significance of DSSC technology can be seen through the number of research articles published annually. FF values depend on *V_oc_* and *J_sc_* , according to Equation (3). PCE values increase with FF increases, according to Equation (4). Therefore, most DSSC publications proved that higher PCE values with TMS-based CE, such as WS_2_, Ag_2_S, CuS, CNT/VS_2_, SnSn_2_/CNTs, FeS_2_ and NiS_n_, compared with Pt-based CE. Figure 9 reveals *V_oc_*_,_ *Vs J_sc_* for both TMS-based CE and Pt-based CE, and demonstrates that *V_oc_* and *J_sc_* values of TMS-based CE are higher than those of Pt-based CE. On the other hand, a slight rise in PV parameter values of Pt-based CE corresponded with some measures of TMS-based CE as shown in Figure 8 and Figure 9.

### 4.2. TMS-Based CEs Applications in QDSSCs

According to Savariraj et al. [126], the surface-active sulfide and disulfide compounds and Cu deficit affected the electrocatalytic activity of Cu_2-x_S thin films. These thin films were utilized in QDSSCs as CEs to reduce polysulfide electrolytes. Temperature-dependent cetyl trimethyl ammonium bromide surfactant determined the preferential adhesion between Cu^2+^ and S^2−^ leading to the specific formation of a Cu_1.8_S stacked platelet-like structure. Therefore, the crab-like Cu-S coordination bond controlled the A/V (area/volume) ratio of Cu_1.8_S thin films and their electrocatalytic performance. The Cu deficit improved the properties of Cu_1.8_S thin films and revealed localized surface plasmon resonance in the near-infrared and excitonic impact in the UV-VIS absorption spectra, which were due to free carriers and the quantum size effect, respectively. A strong PCE of 5.16% was obtained for the film produced at 60 °C by a single-step chemical bath deposition (CBD) approach when these Cu_1.8_S thin films were used as CE in QDSSCs. Based on the observation, Cu_1.8_S was an appropriate and cost-effective replacement for Pt as the CE due to its electrocatalytic properties [126].

Durga et al. [73] described a cost-effective and straightforward low-temperature solution approach to preparing copper sulfide for QDSSCs by using CoS as CE, which exhibited high PCEs of 2.52 and 3.48% at 80 °C for 2 and 3 h, respectively. The enhanced performance of the CoS-3hrs CE was due to the large surface area, good conductivity, and high electrocatalytic activity. Meanwhile, Yuan et al. [127] employed CBD to produce metal sulfides and their composites (CoS/CuS, NiS/CuS, CoS, NiS, and CuS) while utilizing them as CEs of QDSSCs. By investigating the impact of several CEs on cell performance, the CoS/CuS CE demonstrated the best PCE of 5.22%, thus outperforming CuS, CuS/NiS, CoS, and NiS (4.73%, 2.56%, 2.23%, and 1.62%, respectively). The superior electrical conductance and catalytic properties of CuS and CoS contributed to their increased cell efficiency. Additionally, several metal sulfide composites were used to raise *V*_oc_ while maintaining *I*_sc_, which offers promising prospects for improved solar cell capabilities.

Quy et al. [128] produced MoS_2_ films on FTO substrates employing potentiostatic electrodeposition in the island growth mode. As the electrodeposition (ED) time approached 40 min, the MoS_2_ nanoparticle clusters expanded and thickened but still had nanopores separating them. The clusters coalesced for denser films when ED time was increased to 60 min. Compared to other films, the film FTO/MoS_2_ demonstrated significantly increased electrocatalytic activity. This increase was due to the higher electrochemical activity of FTO/MoS_2_, which greatly accelerated charge transport and mass transfer. The QDSSC with FTO/MoS_2_ CE demonstrated an even greater total PCE (3.69%) than Pt CE (2.16%) when used as the CE for QDSSCs and DSSCs. Additionally, the FTO/MoS_2_ CE-equipped DSSC displayed cell efficiency (7.16%) equivalent to the FTO/Pt CE (7.48%). MoS_2_ appeared to be a potential CE material for all DSSCs and QDSSCs.

Quy et al. [90] reported a simple one-step potentiodynamic electrodeposition method to synthesize nickel sulfide (Ni_3_S_4_) films onto FTO substrates. For 4 to 10 cycles, the potential was swept between −0.9 and 0.7 V. The series resistance steadily decreased with increasing Ni_3_S_4_ film thickness, suggesting the metallic conduction of the Ni_3_S_4_ phase. The material possessed a thickness of 110 nm, complete coverage on the FTO substrate, and a unique structure of extremely permeable nanoscale interconnected nanoparticle networks that provide numerous electrochemically active sites to interact with the electrolyte. This Ni_3_S_4_ was deposited for eight cycles and was referred to as FTO/Ni_3_S_4_-8. Compared to FTO/Pt, the film demonstrated strong electrocatalytic activity and high electrochemical resilience in both iodide and polysulfide-based electrolytes. Alternatively, FTO/Ni3S4-10 displayed merging clusters, which resulted in a more compact and porous shape and decreased electrocatalytic activity. The QDSSC was synthesized using FTO/Ni_3_S_4_-8 CE, thus producing an FF of 52.63% and a PCE of 4.57%. A PCE of 8.17% and FF of 68.34% were also attained by a DSSC employing FTO/Ni3S4-8 CE. Moreover, the QDSSC and DSSC with Pt CE achieved PCEs of 2.56 and 7.58%, respectively.

Vijayakumar et al. [129] synthesized a thin manganese cobalt sulfide (MCS) layer on an FTO substrate using a simple electrodeposition approach to fabricate QDSSCs. The developed FTO/MCS films were used as CEs for QDSSCs. Compared to the 1.08% efficiency of Pt CE under one-sun illumination, the QDSSC with the FTO/MCS CE considerably improved PCE to 3.22%. This observation was explained by the binary transition of metal sulfides that significantly improved electrocatalytic activity and electrical conductivity with a connection between FTO and electro-deposited MCS film. Additionally, the FTO/MCS reveals good electrochemical resilience, unlike the traditional Pt CE illustrated in Figure 10.

Li et al. [130] developed ternary spinel MnCo_2_S_4_ that was effectively used as CE for QDSSCs and was anchored to CNTs using a two-step approach of precursor synthesis and ion exchange. According to electrochemical studies, MnCo_2_S_4_ and its composite acquired high catalytic activity for Sn_2_ reduction, as evidenced by the *R*_ct_ values at the interface of MnCo_2_S_4_ (2.86 Ω) and CNTs/MnCo_2_S_4_ (1.09 Ω). With MnCo_2_S_4_ and CNTs/MnCo_2_S_4_ CEs, the PCEs of QDSSCs with CdSe/CdS QD photoanodes reached 2.98 and 4.85%, respectively. The superior PV characteristics of the CNT/MnCo_2_S_4_-based QDSSC were primarily attributed to the synergistic interaction between the outstanding electrocatalytic performance of MnCo_2_S_4_ and the conductance of CNTs. In addition to dramatically shrinking the size of MnCo_2_S_4_ and increasing catalytic activity sites, the inclusion of CNTs also created a crosslinked conductive network that speeds up electron transport. Therefore, CNT/MnCo_2_S_4_ is anticipated to be a reliable CE material for effective QDSSCs owing to the good reducing capability of S_x_^2^ depicted in Figure 11.

Kusuma, Akash and Balakrishna [131] inserted 2D MoS_2_ into the CuS lattice and demonstrated this effective technique for creating more efficient CE material, owing to the synergistic effects of CuS and MoS_2_ in QDSSCs. The SILAR heterojunction-formed device exhibited higher photon absorption. By significantly lowering polysulfide and *R*_ct_ at the CE/electrolyte interface, the higher carrier mobility of 2D MoS_2_ improved kinetics across the interface. The device performed well due to favorable energy level alignment, a wide surface area, and strong lattice matching between the two sulfides. Moreover, the MoS_2_ layers might produce increased catalytic activity due to their abundance of active sites and visible interior edges (pinholes, rips, and flaws) created during hydrothermal reactions.

Tian, Chen and Zhong [132] developed honeycomb-shaped, spherical metallic 1T-MoS_2_ with an easy hydrothermal process and eco-friendly soft templates, demonstrating its effectiveness as a CE for QDSSCs. Their electrochemical experiments produced higher electrocatalytic activity for S_x_^2−^ reduction as the interface *R*_ct_ was only 0.66 Ω with a 3% template. The QDSSCs constructed with Ti-mesh substrate MoS_2_ CEs demonstrated a PCE of 6.03%. This exceptional performance was primarily attributed to the exceptional intrinsic conductance, catalytic performance, hydrophilicity, and unique geometrical advantage of 1T-MoS_2_. A greater number of electrolyte transport channels, active catalytic sites, and material stability improvement were observed when the specific surface area of the honeycomb-shaped 1T-MoS_2_ increased. Based on the experimental findings, 1T-MoS_2_ was predicted to be a competitive CE material for effective QDSSCs.

Table 2 tabulates the different TMS materials synthesized and deposited onto material substrate layers by several researchers, using diverse techniques. The techniques included CBD, potentiostatic and potentiodynamic electrodepositions, hydrothermal methods, and ionic exchange deposition. The PCEs of QDSSCs with TMS-based CEs ranged from 1.62 to 6.70% with the deployment of S^−2^/S_x_^−^ redox-couple electrolytes. The PV parameters, such as FF, *V_oc_*, and *J_sc_*, were equally defined in photoconversion evaluations of the resultant QDSSCs.

As presented in Figure 12, the relative significance of QDSSC technology can be seen through the number of research articles published annually. One can see that most QDSSC publications have higher values of PCE with TMS-based CE such as MoS_2_, CuS, Cu_1.8_S, CoS_n_ and NiS_n,_, compared with Pt-based CE. Figure 13 reveals *V_oc_ Vs J_sc_* for both TMS-based CE and Pt-based CE, which demonstrates that the *V_oc_* and *J_sc_* values of TMS-based CE are higher than those of Pt-based CE leading to higher PCE values due to the corrosion and chemical adsorption by redox polysulfide electrolyte couple onto the surface of Pt CE [126,127,128,129]. 

The power conversion efficiency (PCE) of a quantum dot solar cell (QDSSC) depends on several factors, including the efficiency of the counter electrode. Transition metal counter electrodes have been shown to have higher PCE than platinum counter electrodes in QDSSCs, as shown in Table 2, and this can be explained by several reasons. Higher catalytic activity: Transition metal counter electrodes, such as nickel, cobalt, and iron, have higher catalytic activity than Pt. This means that they can more efficiently catalyze the reduction of the redox electrolyte used in the QDSSC, which is an important step in the generation of electrical current. Lower charge transfer resistance: Transition metal counter electrodes also have lower charge-transfer resistance compared to Pt, which means that they can more easily transfer electrons between the redox electrolyte and the counter electrode. This results in more efficient electron transfer and higher PCE. Lower cost: Transition metals are generally more abundant and less expensive than Pt, which makes them more economically viable as counter electrode materials for QDSSCs. Overall, the combination of higher catalytic activity, lower charge transfer resistance, and lower cost make transition metal counter electrodes a more effective option for QDSSCs, leading to higher power conversion efficiency compared with Pt counter electrodes.

## 5. Conclusions and Perspectives

In conclusion, the field demonstrated incredible advances, as a recent surge in interest in DSSCs and QDSSCs was observed. One of the essential elements in DSSCs and QDSSCs was the CE, which catalyzed the site where regeneration of the redox pair occurred. Traditionally, Pt was preferred for CE as it is a suitable catalyst for redox couple regeneration. However, Pt was expensive, scarce, and subjected to the deterioration of the redox couple. These limitations caused much concern for the potential long-term use of DSSCs and QDSSCs. Hence, developing low-cost, high-efficiency CE catalysts to replace Pt was crucial to increase the competitiveness of DSSCs and QDSSCs, among other solar devices. Based on this review, we note that researchers have recently concentrated their attention on CE catalysts based on TMSs, in which substantial progress was accomplished and successfully demonstrated, as reported in this review.

Various transition metal sulfides, including cobalt sulfide, nickel sulfide, and molybdenum sulfide, have been studied extensively as counter electrodes and have demonstrated excellent electrocatalytic activity, high conductivity, and good stability. Additionally, the synthesis of these materials has become increasingly facile with the development of various synthetic methods, including chemical vapor deposition and solvothermal synthesis.

Given the strong electrocatalytic activity and chemical and mechanical stability towards different redox-couple electrolytes of TMSs, strong capabilities as CEs were observed for TMSs such as Cu_n_S, NiS, and carbonaceous-doped TMS. Thus, efficient and cost-effective alternatives or substitutes were reported when compared to Pt. In addition to single-metal sulfides, researchers have also developed TMS composites to obtain additional benefits and improved characteristics. Conversely, the primary issues with this kind of catalyst were the high energy consumption and toxic gas emitted during the synthesis processes. For this type of CE catalyst, it was crucial to explore alternative synthesis approaches using environmentally friendly technology and reduced energy usage. Carbon-based materials have been considered promising substitutes for capital-intensive Pt materials as CEs in DSSCs and QDSSCs. They are associated with highly desirable properties, including high thermal stability, high electrolyte reduction reactivity, strong resistance to electrode corrosion, high electrical conductivity, and high catalytic activity. These carbon-based CEs could equally be subjected to pre- and post-treatment processes to evaluate their responses following the interaction with non-liquid phased electrolytes. Such electrolytes could include gel, solid, and quasi-solid electrolytes.

However, further research is needed to optimize the performance of these materials and fully understand their underlying mechanisms. In particular, the impact of factors such as morphology, crystal structure, and dopants on the electrocatalytic properties of transition metal sulfides needs to be thoroughly investigated. Moreover, the integration of transition metal sulfides into larger scale photovoltaic devices needs to be explored, as well as the potential for these materials to be used in other electrochemical applications beyond solar cells. Overall, the use of transition metal sulfides as counter electrodes in dye-sensitized and quantum dot-sensitized solar cells is a promising avenue for improving the efficiency and reducing the cost of these important renewable energy technologies.

## Figures and Tables

**Figure 1 materials-16-02881-f001:**
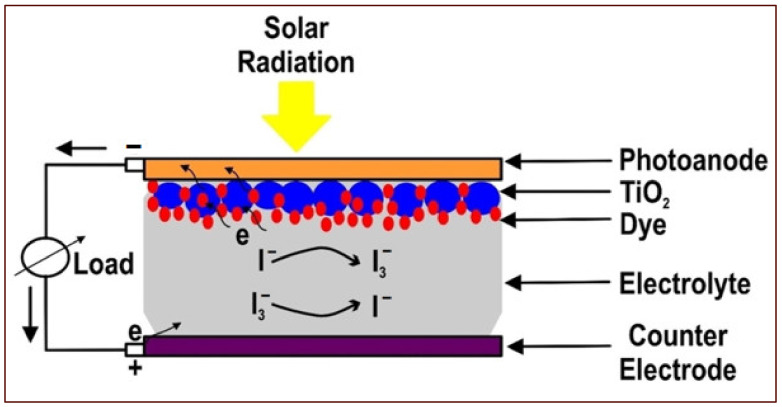
Typical representation of DSSC showing the main components, comprising a photoanode, a semiconducting oxide layer (TiO_2_), molecular dye, a redox-couple electrolyte, and counter electrode. Electrons exit and make re-entry through the photoanode and counter-electrode systems, respectively. Modified after reference [36].

**Figure 2 materials-16-02881-f002:**
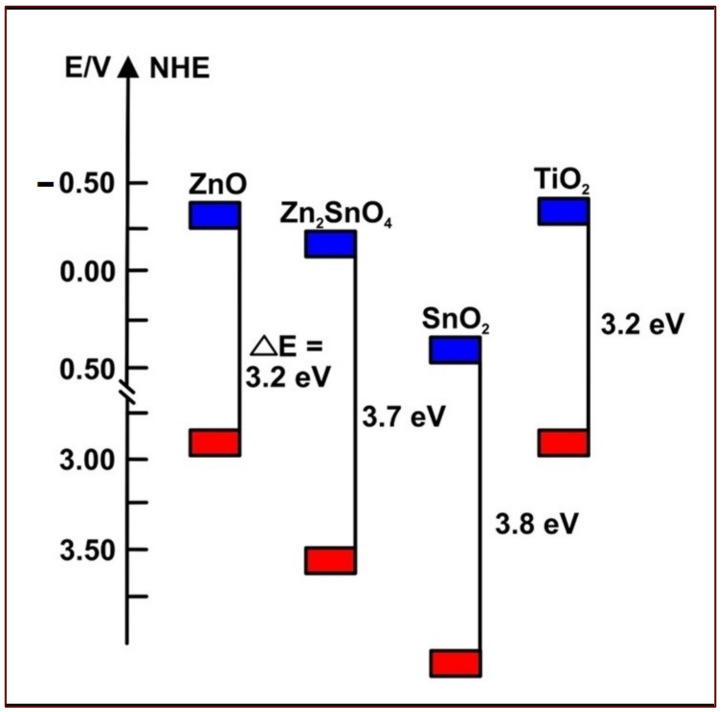
The energy band positions of commonly used semiconductors indicate ZnO as a promising alternative to TiO_2,_ given their matching energy band structures. Modified after reference [34].

**Figure 3 materials-16-02881-f003:**
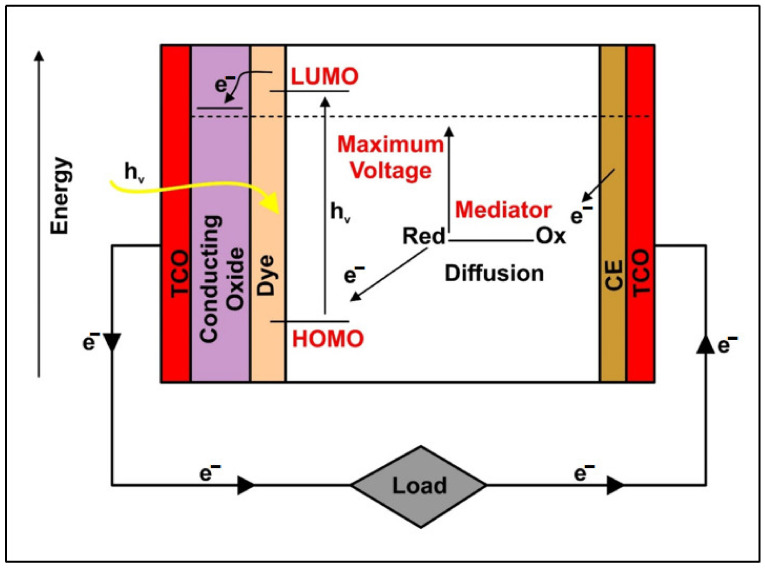
Schematic representation of the DSSC working principle with indications of electron movements from the HOMO to LUMO regions of the cell and passing through the external circuitry system for accumulation at the CE.

**Figure 4 materials-16-02881-f004:**
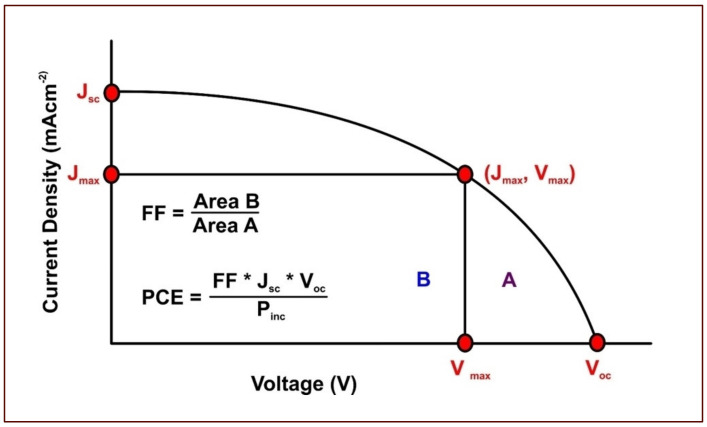
Current–voltage curve for evaluating DSSC performance using the cell parameters. Modified after reference [37].

**Figure 5 materials-16-02881-f005:**
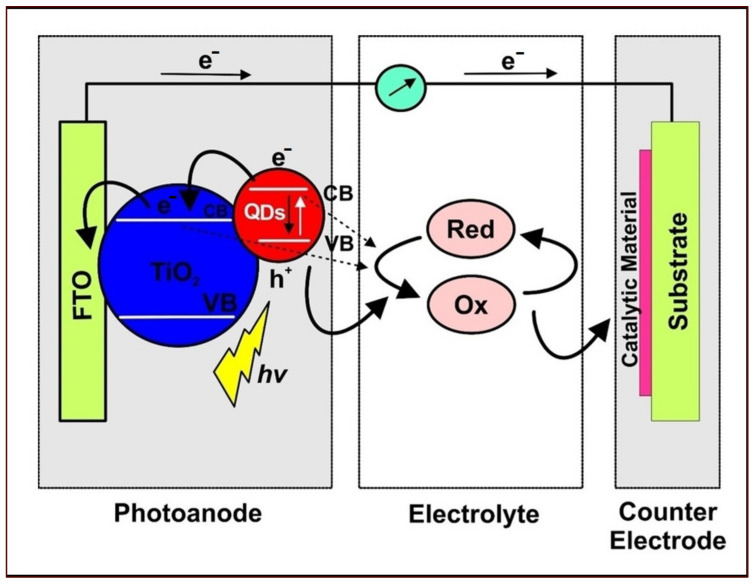
Schematic representation of a typical QDSSC and its associated working principle. Modified after reference [13].

**Figure 6 materials-16-02881-f006:**
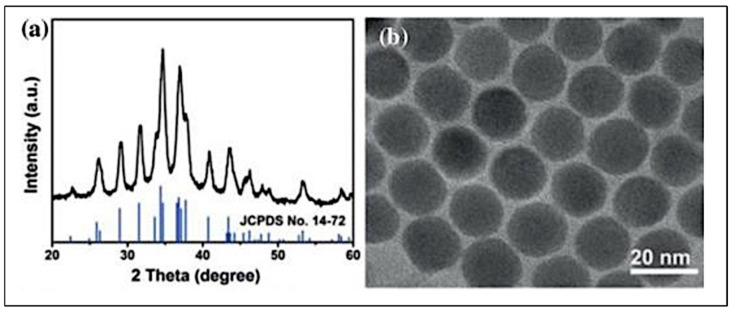
(**a**) XRD of the synthesized product and standard Ag_2_S (JCPDS No. 14–72) in black and blue line, respectively. and (**b**) TEM of Ag_2_S nano- crystals. Adapted with permission from reference [108]. Copyright 2022, Wiley Publishing.

**Figure 7 materials-16-02881-f007:**
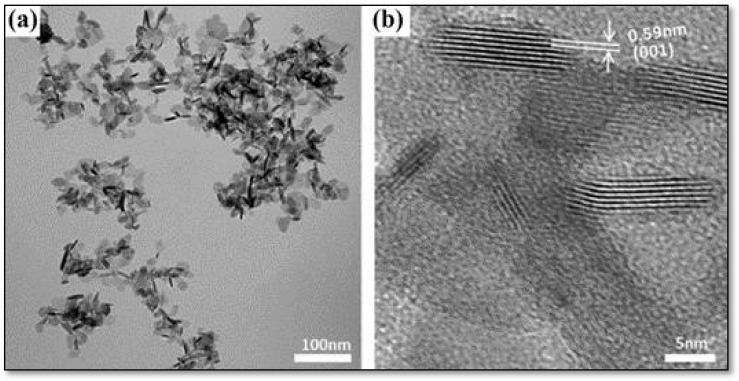
(**a**) TEM and (**b**) HRTEM images of as-prepared SnS2 nanosheets. Adapted with permission from [112]. Copyright 2022, Wiley Publishing.

**Figure 8 materials-16-02881-f008:**
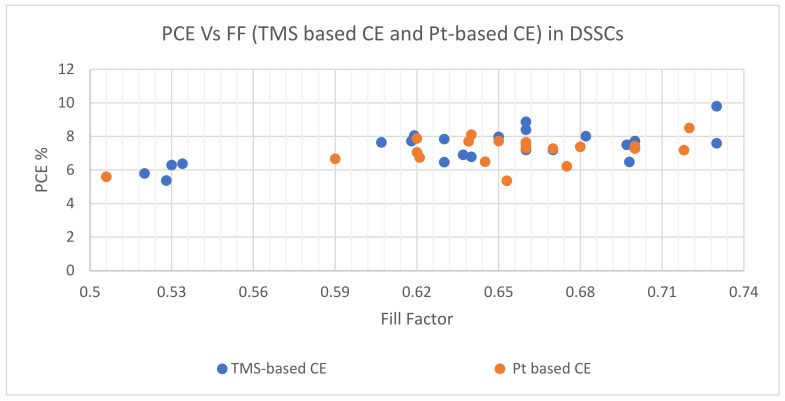
The PCE Vs FF performance for TMS-based CE and Pt-based CE in DSSCs.

**Figure 9 materials-16-02881-f009:**
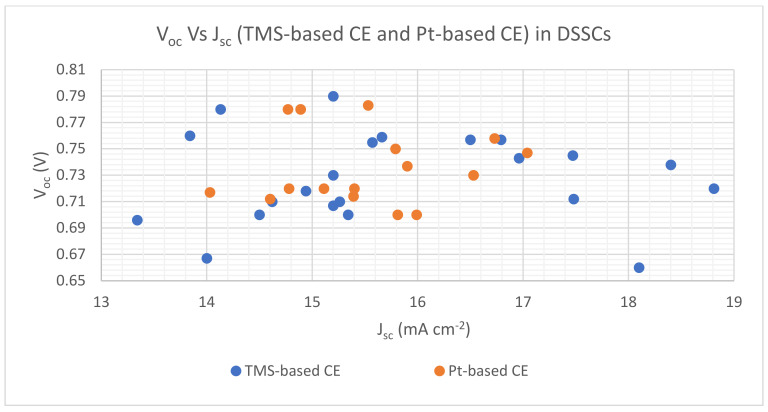
The V_oc_ Vs J_sc_ for TMS-based CE and Pt-based CE in DSSCs.

**Figure 10 materials-16-02881-f010:**
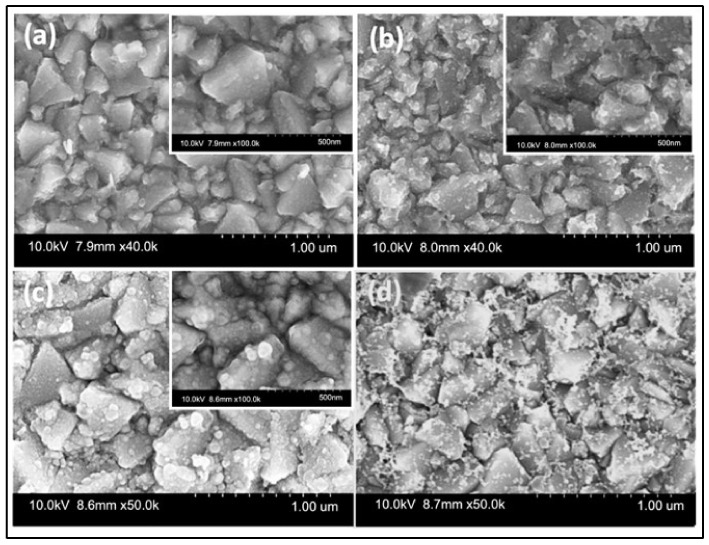
FESEM snaps of (**a**–**c**) FTO/MCS (3, 6, and 9 cycles) and (**d**) FTO/Pt CEs. Adapted with permission from reference [129]. Copyright 2022, IOP Publishing.

**Figure 11 materials-16-02881-f011:**
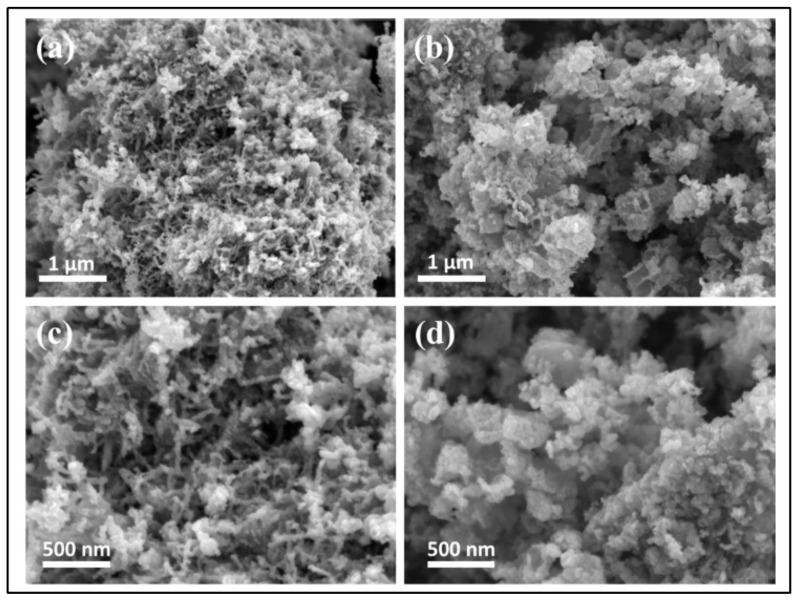
SEM of (**a**,**c**) CNTs/MnCo_2_S_4_ and (**b**,**d**) MnCo_2_S_4_. Adapted with permission from [130] Copyright 2022, American Chemical Society.

**Figure 12 materials-16-02881-f012:**
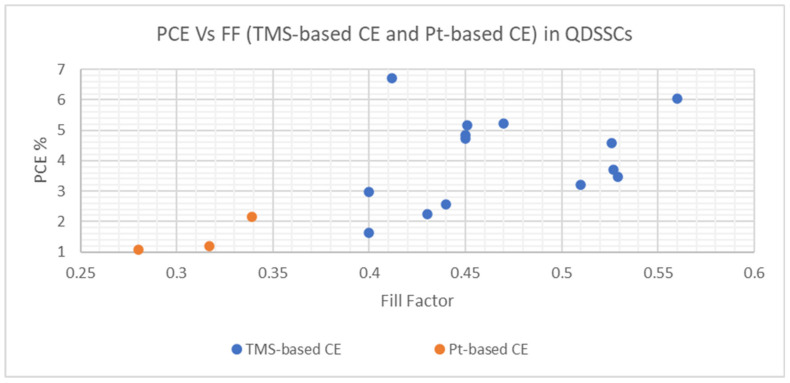
The PCE Vs FF performance for TMS-based CE and Pt-based CE in QDSSCs.

**Figure 13 materials-16-02881-f013:**
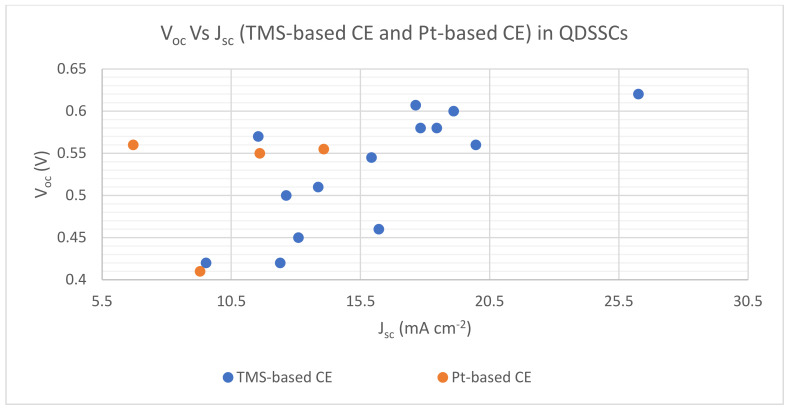
The V_oc_ Vs J_sc_ for TMS-based CE and Pt-based CE in QDSSCs.

**Table 1 materials-16-02881-t001:** Summary of 0TMS-based CEs synthesized and applied in DSSCs. The resulting PV performance (PCE, FF, *V*_oc_, and *J*_sc_) values of DSSCs with TMS-based CEs are compared to those of DSSCs with Pt-based CEs.

TMS Based CE	Synthesis and Deposition Method	PCE Performance (%)		FF		*V*_oc_ (V)		*J*_sc_ (mA cm^−2^)		Electrolyte	Comments on PV Performance of DSSCs with Different TMS-Based CEs	Ref.
		TMS-Based CE	Pt Based CE	TMS-Based CE	Pt Based CE	TMS-Based CE	Pt Based CE	TMS-Based CE	Pt Based CE			
WS_2_	Simple chemical method	7.73	7.64	0.70	0.66	0.78	0.78	14.13	14.89	I^−^/I_3_^−^ redox-couples	Comparable PCEs for DSSCs with WS_2_, MoS_2,_ and platinized CEs	[107]
MoS_2_		7.59	7.64	0.73	0.66	0.76	0.78	13.84	14.89			
Ag_2_S nanoparticles (NPs)	Colloidal synthesis process	8.40	8.11	0.66	0.64	0.757	0.758	16.79	16.73	I^−^/I_3_^−^ redox-couple	Higher PCE for DSSC with Ag_2_S NPs CE in comparison to that with platinized CE	[108]
CuS nanosheet (NS)	Electrospinning	6.38	5.60	0.534	0.506	0.66	0.70	18.10	15.81	I^−^/I_3_^−^ redox-couple	Higher PCE for DSSC with CuS NS CE in comparison to DSSC with Pt CE	[109]
Sb_2_S_3_ film	Hydrothermal and post-annealing treatments	5.37	5.36	0.528	0.653	0.70	0.65	14.5	12.5	I^−^/I_3_^−^ redox-couple	Associated PCE of DSSC with Sb_2_S_3_ CE was higher than Pt CE	[110]
CNTs/VS_2_	Hydrothermal method	8.02	6.49	0.682	0.645	0.755	0.717	15.57	14.03	I^−^/I_3_^−^ redox-couple	DSSC with CNTs/VS_2_ CE showed higher conductivity, better electrocatalytic activity, and higher PCE compared with Pt CE	[111]
SnS_2_ nanosheet (NS)	Solution-processed approach	7.64	7.71	0.607	0.639	0.743	0.730	16.96	16.53	I^−^/I_3_^−^ redox-couple	An increase in PCE was recorded from 7.64% (DSSC with SnS_2_ NS CE) to 8.06% for DSSC with SnS_2_ NS + CNPs CE	[112]
SnS_2_ NS + carbon nanoparticles (CNPs)	Solution-based approach	8.06	7.71	0.619	0.639	0.745	0.730	17.47	16.53			
SnS_2_ NPs	Hydrothermal method	6.30	6.67	0.53	0.59	0.759	0.783	15.66	15.53	I^−^/I_3_^−^ redox-couple	PCEs were comparable to the DSSCs, given the closeness of associated values	[113]
Co_3_S_4_ NSs	Hydrothermal method	7.19	7.27	0.66	0.67	0.70	0.70	15.34	15.99	I^−^/I_3_^−^ redox-couple	DSSC with Co_3_S_4_ NSs CE indicated comparable PCE to platinized CE	[114]
CoS_2_ nanocrystals	Hydrothermal method	6.78	7.38	0.64	0.68	0.71	0.72	14.62	14.78	I^−^/I_3_^−^ redox-couple	DSSC with CoS_2_ CE exhibited PCE comparable to Pt CE	[115]
CoS film	Electrophoretic deposition and ion exchange deposition	7.72	7.18	0.618	0.718	0.757	0.792	16.50	12.63	I^−^/I_3_^−^ redox-couple	PCEs of the DSSCs with CoS film and Pt CEs were relatively comparable, given the low cost of CoS film, would be more suitable for application	[116]
FeS-HEF	Solution-phase chemical method	8.88	7.73	0.66	0.65	0.72	0.75	18.81	15.79	I^−^/I_3_^−^ redox-couple	DSSC with FeS-HEF CE demonstrated excellent electrocatalytic activity and produced PCE higher than Pt CE	[60]
FeS_2_ film	Spray pyrolysis	7.97	7.54	0.65	0.66	0.79	0.78	15.20	14.77	I^−^/I_3_^−^ redox-couple	FeS_2_ CE associated PCE was higher than PCE (Pt) of 7.54% with the use of I^−^/I_3_^−^ redox couples	[117]
FeS nanorods (NRs) (FeS NRs)	Electrospinning	6.47	7.05	0.63	0.62	0.667	0.714	14.00	15.39	I^−^/I_3_^−^ redox couple	PCE of DSSC with FeS NRs CE was comparable to Pt CE	[118]
MoS_2_	Chemical vapor deposition	7.50	7.28	0.697	0.700	0.707	0.712	15.2	14.6	I^−^/I_3_^−^ redox couple	DSSC with MoS_2_ CE produced higher PCE in comparison to Pt CE, producing PCE of 7.28%	[119]
MoS_2_ with graphite paper (GP) as TCO	Solution-processed route	6.48	6.22	0.698	0.675	0.696	0.720	13.34	12.79	I^−^/I_3_^−^ redox couple	DSSC with MoS_2_ CE outperformed Pt CE with PCE of 6.22%	[120]
MoS_2_	Heat-sintering method with a near-infrared (IR) pulsed laser	7.19	7.42	0.67	0.70	0.718	0741	14.94	14.30	I^−^/I_3_^−^ redox couple	DSSC with laser-sintered MoS_2_ CE exhibited good electrocatalytic performance, with its PCE comparable to DSSC with Pt CE	[121]
MoS_2_ film	Solid state sulfurization method	5.80	7.30	0.52	0.66	0.73	0.72	15.20	15.40	I^−^/I_3_^−^ redox couple	PCE of DSSC with patterned MoS_2_ CE was lower but comparable to Pt CE	[122]
NiS NTs	Electrochemical deposition	9.80	8.50	0.73	0.72	0.738	0.737	18.40	15.90	I^−^/I_3_^−^ redox couple	DSSC with NiS NTs CE demonstrated both excellent electrocatalytic activity towards I_3_^–^ reduction and high electrochemical stability, resulting in higher PCE	[123]
NiS_2_ hierarchical hollow microspheres	Hydrothermal method	7.84	7.89	0.63	0.62	0.712	0.747	17.48	17.04	I^−^/I_3_^−^ redox couple	DSSC with NiS_2_ CE demonstrated excellent electrochemical catalytic activity, and associated PCE was comparable to Pt CE	[124]
NiS hollow spheres	Hydrothermal method	6.90	6.75	0.637	0.621	0.71	0.72	15.26	15.11	I^−^/I_3_^−^ redox couple	DSSC with hollow NiS sphere CE exhibited better electrochemical catalytic activity, as confirmed by its higher PCE	[125]

**Table 2 materials-16-02881-t002:** Summary of TMS-based CEs synthesized and applied in QDSSCs. The resulting PV performances (PCE, FF, *V*_oc_, and *J*_sc_) values of QDSSCs with TMS-based CEs are compared with those of QDSSCs with Pt-based CEs.

TMS Based CE	Synthesis and Deposition Method	PCE Performance (%)		FF		*V*_oc_ (V)		*J*_sc_ (mA cm^−2^)		Electrolyte	Comments on PV Performance of QDSSCs with Different TMS-Based CEs	Ref.
		TMS-Based CE	Pt-Based CE	TMS-Based CE	Pt Based CE	TMS-Based CE	Pt Based CE	TMS-Based CE	Pt Based CE			
Cu_1.8_S nanoplates	Chemical bath deposition method	5.16	1.19	0.451	0.317	0.60	0.56	19.10	6.70	S^−2^/S_x_^−^ redox-couple	QDSSC with Cu_1.8_S nanoplates CE exhibited the best photoconversion behavior in comparison with platinized CE	[126]
CoS leaf-like nanostructure	Solution-based approach	3.48	-	0.529	-	0.57	-	11.54	-	S^−2^/S_x_^−^ redox-couple	72.41% increase in PCE of QDSSC resulting from a 2 to 3 h heat treatment process of CoS leaf-like nanostructure	[73]
CoS/CuS	CEs were deposited onto FTO substrates by chemical bath deposition (CBD)	5.22	-	0.47	-	0.56	-	19.96	-	S^−2^/S_x_^−^ redox-couple	The utilization of different TMSs and their composites as CEs indicated the variation of PCEs for QDSSCs from 1.62 to 5.22%	[127]
CuS		4.73	-	0.45	-	0.58	-	17.82	-			
CuS/NiS		2.56	-	0.44	-	0.45	-	13.09	-			
CoS		2.23	-	0.43	-	0.42	-	12.39	-			
NiS		1.62	-	0.40	-	0.42	-	9.52	-			
MoS_2_	CE was deposited onto the FTO substrate by potentiostatic electrodeposition	3.69	2.16	0.527	0.339	0.51	0.55	13.86	11.6	S^−2^/S_x_^−^ redox-couple	QDSSC with MoS_2_ CE exhibited a much higher PCE than platinized CE	[128]
Ni_3_S_4_ film	Potentiodynamic electrodeposition	4.57	2.56	0.526	0.328	0.545	0.555	15.92	14.07	S^−2^/S_x_^−^ redox-couple	QDSSC with Ni_3_S_4_ film CE exhibited better photoconversion behavior in comparison to Pt CE	[90]
Manganese cobalt sulfide (MCS) thin film	Electrochemical Synthesis, CE was deposited on FTO by CBD	3.22	1.08	0.51	0.28	0.50	0.41	12.62	9.28	S^−2^/S_x_^−^ redox-couple	QDSSC with Pt CE resulted in poor FF and much lower PCE of 1.08% in comparison to MCS thin film CE	[129]
Ternary spinel MnCo_2_S_4_	Ionic exchange deposition, CEs were deposited onto FTO substrates by drop-coating.	2.98	-	0.40	-	0.46	-	16.20	-	S^−2^/S_x_^−^ redox-couple	An improvement in PCE was identified with the utilization of MnCo2S4/CNT as CE in QDSSC	[130]
Carbon nanotubes (CNTS)/MnCo_2_S_4_		4.85	-	0.45	-	0.58	-	18.45	-			
MoS_2_/CuS nanohybrid	Hydrothermal method	6.70	-	0.412	-	0.62	-	26.25	-	S^−2^/S_x_^−^ redox-couple	Reported values demonstrated good PV performance and were based on the statistical average of six cells	[131]
Honeycomb spherical metallic 1T-MoS_2_	Hydrothermal method	6.03	-	0.56	-	0.607	-	17.63	-	S^−2^/S_x_^−^ redox-couple	QDSSC with 1T-MoS_2_ CE demonstrated good photo conversion efficiency supported by associated parametric values	[132]

## Data Availability

Not applicable.

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
