# Peer review of "A Review of Transition Metal Sulfides as Counter Electrodes for Dye-Sensitized and Quantum Dot-Sensitized Solar Cells"

_materials, 2023, doi:10.3390/ma16072881_

Round 1

Reviewer 1 Report

Authors summarize advance on the third-generation solar cells including dye-sensitised solarcells (DSSCs) and quantum dot-sensitised solar cells (QDSSCs) which TMSs replace Pt for CEs, and point out the future direction of TMS-based CEs toward performance efficiency improvement of DSSCs and QDSSCs in the most cost-effective and environmentally-friendly manner. It is an interesting subject, so I agree it to be published after revised.

(1) Authors should point out whether similar subject have been summarized, or summarized stage;

(2) Authors revised some errors of words and grammar;

(3) Authors noted PPCE in line 3 on the page 18;

Authors should point out what materials are suitable to CEs in TMSs. 

Author Response

Dear Sirs

Thank you for the opportunity to revise our paper, “A Review of Transition Metal Sulfides as Counter Electrodes for Dye-Sensitized and Quantum Dot-Sensitized Solar Cells” (2256380). We have noted the comments from you and the reviewers and have made changes to our paper in response. We believe these changes have greatly improved the paper and we hope it is now acceptable for publication in Materials.

The changes we have made are outlined below and can be seen in tracked changes in the attached copy of the revised paper.

Comments

Corrections

1. Authors should point out whether similar subject have been summarized, or summarized stage

 This review paper is providing an overview of the current state of research on a similar topic. It is highlighted in section 1.0 in the last paragraph.

2. Authors revised some errors of words and grammar;

 Errors in writing and grammar are corrected accordingly

3. Authors noted PPCE in line 3 on page 18;

Corrected ( PCE),  line 3 page 19

4. Authors should point out what materials are suitable for CEs in TMSs. 

1. Section 4.1 Line 35 on page 17

2. Section 4.2 Line 13 on page 27

3. Conclusion and Perspectives Line 17 page 32

Thank you again for the opportunity to improve our paper and the suggestions for doing so.

(Attach the ‘showing changes’ and ‘changes incorporated’ files, clearly distinguished by their file names, e.g., ‘… copy with tracked changes and ‘… copy proposed for publication)

Reviewer 2 Report

This manuscript has reviewed the advance of transition metal sulfides as counter electrodes for dye-sensitized and quantum dot-sensitized Solar Cells. I support the publication of this work in materials after the following revisions.

1. Introduction section should be shortened, especially, paragraph 1 to 5 in introduction section. The contents about popularization of science should be simplified.

2. The literatures were introduced separately in Section 4.1 & 4.2, lack of the corresponding analysis and classification. It should be noted that TMSs is a big family.

3. Concise conclusion and analysis or relation is necessary in every sections.

4. The section of Conclusion and Perspectives was poorly organized, and a major revision is necessary, especially, more contents about outlook or perspectives should be provided.

5. Some very related literatures should be incited to enrich the contend in this review, such as Adv. Mater., 2016, 28, 1917-1933; Chem. Soc. Rev., 2015, 44, 2702-2712; Electrochem. Energ. Rev., 2021, 4, 194-218; Chinese Chemical Letters 34 (2023) 107119.

6. The english writting should be further polished.

Author Response

Dear Sirs

Thank you for the opportunity to revise our paper, “A Review of Transition Metal Sulfides as Counter Electrodes for Dye-Sensitized and Quantum Dot-Sensitized Solar Cells” (2256380). We have noted the comments from you and the reviewers and have made changes to our paper in response. We believe these changes have greatly improved the paper and we hope it is now acceptable for publication in Materials.

The changes we have made are outlined below and can be seen in tracked changes in the attached copy of the revised paper.

Comments

Corrections

1. Introduction section should be shortened, especially, paragraph 1 to 5 in introduction section. The contents about popularization of science should be simplified

Some paragraph in the Introduction section is revised and combined with a shorter version and the repeated details are removed.

2. The literatures were introduced separately in Section 4.1 & 4.2, lack of the corresponding analysis and classification. It should be noted that TMSs is a big family.

The collected data summarized in Table 1 for Section 4.1 and  Table 2 for Section 4.2, are analysed and discussed as represented by Figure 8 and Figure 9 for DSSCs and  Figure 12 and Figure 13 for QDSSCs in the respective sections.

3. Concise conclusion and analysis or relation is necessary in every sections.

4. The section of Conclusion and Perspectives was poorly organized, and a major revision is necessary, especially, more contents about outlook or perspectives should be provided.

The section has been improved to include comments on lines 17, 27 and 48 on page 32.

5. Some very related literatures should be incited to enrich the contend in this review, such as Adv. Mater., 2016, 28, 1917-1933; Chem. Soc. Rev., 2015, 44, 2702-2712; Electrochem. Energ. Rev., 2021, 4, 194-218; Chinese Chemical Letters 34 (2023) 107119.

This article:  Chem. Soc. Rev., 2015, 44, 2702-2712 is related to our topic.

It has been added in the last paragraph of the Introduction section (Reference no 33)

6. The English writing should be further polished

Errors in writing and grammar are corrected accordingly

Thank you again for the opportunity to improve our paper and the suggestions for doing so.

(Attach the ‘showing changes’ and ‘changes incorporated’ files, clearly distinguished by their file names, e.g., ‘… copy with tracked changes and ‘… copy proposed for publication)

Reviewer 3 Report

In this review, the author presented the major components and working principles dye-sensitized solar cells (DSSCs) and quantum dot-sensitized solar cells (QDSSCs). The review is of considerable interest and well done. I recommend it to be published after a minor revision.

1. The novelty needs to refinement and should be highlighted in the introduction part.

2. The manuscript contains some minor typo/grammar errors, please check all of it.

3. The author should better improve the beauty and quality of the figures in the manuscript.

4.Some publications are suggested to refer to improve the quality of the manuscript, such as: https://doi.org/10.1016/j.ceramint.2022.10.029, https://doi.org/10.1016/j.est.2023.106806.

5. Abstract not targeted; the authors should rephrase it.

Author Response

Dear Sirs

Thank you for the opportunity to revise our paper, “A Review of Transition Metal Sulfides as Counter Electrodes for Dye-Sensitized and Quantum Dot-Sensitized Solar Cells” (2256380). We have noted the comments from you and the reviewers and have made changes to our paper in response. We believe these changes have greatly improved the paper and we hope it is now acceptable for publication in Materials.

The changes we have made are outlined below and can be seen in tracked changes in the attached copy of the revised paper.

Thank you again for the opportunity to improve our paper and the suggestions for doing so.

Comments

Corrections

1. The novelty needs to refinement and should be highlighted in the introduction part.

Previous review articles have reported on the current performance of DSSCs and QDSSC in general without focusing on both transition metal chalcogenides (TMCs) and transition metal dichalcogenides (TMDs) [33]  In this review, the current performance of electrochemical and photovoltaic properties of low-cost catalytic CEs developed from earth-abundant TMCs including TMDs and their composites with other materials are discussed which is the novelty of this review paper.

2. The manuscript contains some minor typo/grammar errors, please check all of it.

Errors in writing and grammar are corrected accordingly

3. The author should better improve the beauty and quality of the figures in the manuscript.

The figures have been replaced with clearer versions.

4. Some publications are suggested to refer to improve the quality of the manuscript, such as: https://doi.org/10.1016/j.ceramint.2022.10.029, https://doi.org/10.1016/j.est.2023.106806.

 Since this review paper will focus on the counter electrode application in third-generation solar cells, especially the dye and quantum dots-sensitized solar cells, we have to exclude other applications such as supercapacitors. This is because our secondary data is based on the value of PCE, FF, Jsc and Voc (Power conversion efficiency, fill factor, short-circuit current density and Open circuit voltage) that are obtained from the primary data of the articles related to the solar application

5. Abstract not targeted; the authors should rephrase it.

 The abstract has been revised to include a clear objective and target as shown in line 7, page 1.